# A Multimodel Study of the Role of Novel PKC Isoforms in the DNA Integrity Checkpoint

**DOI:** 10.3390/ijms242115796

**Published:** 2023-10-31

**Authors:** Sara Saiz-Baggetto, Laura Dolz-Edo, Ester Méndez, Pau García-Bolufer, Miquel Marí, M. Carmen Bañó, Isabel Fariñas, José Manuel Morante-Redolat, J. Carlos Igual, Inma Quilis

**Affiliations:** 1Departament de Bioquímica i Biologia Molecular, Universitat de València, 46100 Burjassot, Spain; sara.saiz@uv.es (S.S.-B.); laura.dolz-edo@uv.es (L.D.-E.); bano@uv.es (M.C.B.); 2Institut de Biotecnologia i Biomedicina (BIOTECMED), Universitat de València, 46100 Burjassot, Spainisabel.farinas@uv.es (I.F.); jm.morante@uv.es (J.M.M.-R.); 3Departament de Biologia Cellular, Biologia Funcional i Antropologia Física, Universitat de València, 46100 Burjassot, Spain

**Keywords:** PKC, DNA integrity checkpoint, *S. cerevisiae*, mESCs

## Abstract

The protein kinase C (PKC) family plays important regulatory roles in numerous cellular processes. *Saccharomyces cerevisiae* contains a single PKC, Pkc1, whereas in mammals, the PKC family comprises nine isoforms. Both Pkc1 and the novel isoform PKCδ are involved in the control of DNA integrity checkpoint activation, demonstrating that this mechanism is conserved from yeast to mammals. To explore the function of PKCδ in a non-tumor cell line, we employed CRISPR-Cas9 technology to obtain PKCδ knocked-out mouse embryonic stem cells (mESCs). This model demonstrated that the absence of PKCδ reduced the activation of the effector kinase CHK1, although it suggested that other isoform(s) might contribute to this function. Therefore, we used yeast to study the ability of each single PKC isoform to activate the DNA integrity checkpoint. Our analysis identified that PKCθ, the closest isoform to PKCδ, was also able to perform this function, although with less efficiency. Then, by generating truncated and mutant versions in key residues, we uncovered differences between the activation mechanisms of PKCδ and PKCθ and identified their essential domains. Our work strongly supports the role of PKC as a key player in the DNA integrity checkpoint pathway and highlights the advantages of combining distinct research models.

## 1. Introduction

Cell division is a central biological process. The checkpoints are mechanisms of control that ensure the order of events in the cell cycle and the cell survival in the face of cell disturbances during division [1]. There are several checkpoint mechanisms that operate in different phases of the cell cycle in response to different stimuli: the spindle checkpoint, which prevents errors in chromosome segregation by blocking the metaphase–anaphase transition [2,3,4]; different checkpoints that respond to defects in morphogenesis and the cell wall [5,6,7,8,9] and the DNA integrity checkpoint, that includes the DNA replication checkpoint and the DNA damage checkpoint, which responds to any DNA alterations and coordinates the cell cycle progression with the repair of damaged DNA [10,11,12]. The maintenance of genome integrity is an essential aspect of cell physiology since cells are continuously exposed to a high number of DNA lesions, caused by endogenous and environmental agents, which can compromise the information contained in the genetic material [13].

Proteins involved in the DNA integrity checkpoint were originally identified because their loss of function results in defective cell cycle arrest in response to genotoxic agent exposure [14]. Failures in these checkpoint controls are dramatically associated with a decrease in cell resistance to genotoxic stress, defects in cell cycle arrest when DNA is damaged, and genomic instability, which compromise cell viability. In multicellular organisms, this can lead to the development of cancer [15,16,17]. The DNA integrity checkpoint implies a set of proteins with functional counterparts in all eukaryotic cells, since checkpoint mechanisms are very well conserved throughout evolution. Proteins with kinase activity play a central role in the response to DNA damage. The main components of the DNA integrity checkpoint are the sensor or apical kinases, Mec1 and Tel1 in *S. cerevisiae*, ATR and ATM in mammals. These are responsible for detecting DNA damage and, through the phosphorylation of a series of intermediary proteins, transducing the signal to the effector kinases, Chk1 and Rad53 in yeast, and CHK1 and CHK2 in mammals. Activation of these kinases results in transient cell cycle arrest, activation of transcriptional programs, DNA repair, or if the damage is too severe, cell senescence or programmed cell death [11]. The perception that checkpoint kinases operate in a linear signaling pathway is, however, a simplified view of the DNA damage response, which has evolved into a broader view, thanks to insights gained from numerous global phosphoproteomics experiments, in which these kinases act in a complex network involving hundreds of substrates [12].

The term protein kinase C (PKC) defines a group of Ser/Thr phospholipid-dependent kinases that regulate a wide variety of cellular functions, such as cell proliferation, differentiation, and apoptosis [18,19]. In mammals, 12 different isoenzymes constitute the PKC superfamily. All of these isoenzymes have different biochemical properties, tissue-specific distributions, and subcellular localizations, which give them a variety of functions and broad specificity [18,19]. Consequently, the absence or activation of different isoforms can have pleiotropic effects on the cells. In fact, alterations in PKC signaling have been associated with multiple human diseases, including the development and progression of cancer [20,21]. The mammalian PKC superfamily is part of the AGC group of protein kinases (named after the protein kinase A, G, and C families), a group of cytoplasmic Ser/Thr kinases that share an activation mechanism based on the modification of interactions between the C-terminal kinase domain and the N-terminal regulatory domain, either through intra or intermolecular changes or its phosphorylation [22,23]. The mammalian PKC superfamily is divided into two groups: the PKC family itself (nine isoenzymes) and the group formed by PKC-related kinases, called PRKs or PKNs (three isoenzymes: PRK1, PRK2, and PRK3). The PKC family isoforms are classified into three groups according to their structural domains and regulatory mechanisms [19,21,24,25]. The first group is formed by classical or conventional PKCs (cPKCs), includes the α, β, and γ isoforms, and is capable of binding diacylglycerol (DAG) and phorbol esters through the C1 domain formed by two repetitions of the zinc finger motif. The presence of a C2 motif with five Asp residues allows cPKCs to be sensitive to Ca^2+^ regulation. The second group is formed by novel PKCs (nPKCs) and includes the δ, ε, η, and θ isoforms. nPKCs also have a C1 domain which binds DAG and phorbol esters, but cannot bind Ca^2+^, due to the existence of a C2-like motif that lacks Asp residues necessary for the coordination of the cation. The third group formed by atypical PKCs (aPKCs) consists of ζ and λ/ι (mouse/human) isoforms. These isoforms lack a C2 domain, making them insensitive to Ca^2+^ regulation, as well as one of the zinc finger repeats in their C1 domain, which also makes them insensitive to DAG and phorbol esters. However, the presence of the PB1 domain facilitates the interaction of aPKCs with scaffold proteins, which allows their activation [26,27,28]. All domains that constitute these isoforms are connected to each other through highly flexible binding regions, among which the hinge region is the most remarkable, which directly connects the regulatory region with the catalytic region. The catalytic fragment, which includes the kinase domain and C-terminal tail, contains three phosphorylation sites relevant to the stability of the enzyme and its interaction with other kinases: one in the activation loop, one in the turn motif, and one in the hydrophobic motif. All isoforms of the PKC family have a pseudosubstrate sequence in the regulatory region, whose interaction with the catalytic region maintains the enzyme in an inactive folded conformation. The *S. cerevisiae* genome encodes a single enzyme homologous to mammalian PKC named Pkc1 [29]. *PKC1* deletion is lethal under normal growth conditions due to cell lysis resulting from defects in cell wall integrity, as the viability of the *pkc1* mutant can be rescued in the presence of osmotic stabilizers such as sorbitol [30,31]. Pkc1 is a 1151 amino acid protein whose structure is divided into an N-terminal region containing the HR1, C1, and C2 regulatory domains and a C-terminal region containing the kinase domain, separated by the hinge region. In addition, Pkc1 has a pseudosubstrate sequence that inhibits enzyme activity by interacting with its catalytic domain in the absence of cell wall stress [32]. Due to Pkc1 structural characteristics, since it has all the regulatory domains present in mammalian PKCs although some of them contain modifications or mutations in key residues that prevent them from being functional, Pkc1 has been considered an archetypal PKC [24,32,33,34,35].

Yeast Pkc1 has been related to DNA metabolism [8]. In particular, the *pkc1* mutant presents a high rate of mitotic recombination, a representative characteristic of genomic integrity defects [36]. Pkc1 phosphorylates and activates CTP synthetase [37,38] and, consistent with the role of Pkc1 in nucleotide biosynthesis, phosphorylation of the Rnr2 and Rnr4 subunits of RNR has been detected in a proteomic assay when overexpressing a hyperactive allele of Pkc1 [39]. On the other hand, the *pkc1* mutant is hypersensitive to genotoxic agents, such as methyl methanesulfonate (MMS), hydroxyurea (HU), or bleomycin [40,41,42,43]. Supporting the role of Pkc1 in the cellular response to genotoxic stress, it has been described that Pkc1 is phosphorylated in response to DNA damage, and this phosphorylation is mediated by checkpoint sensor kinases [43,44] and, more importantly, that both Pkc1 and its catalytic activity are necessary for the correct activation of the DNA integrity checkpoint [8,43]. The fact that mutants in the MAPK cascade efficiently activate the response to DNA damage [43,45] indicates that Pkc1 regulates the checkpoint independently of the cell wall integrity pathway, the main signaling pathway in which it is involved. In our group, we also established a connection of the novel mammalian isoform PKCδ with the response to DNA damage [8,43]. We described that in yeast, it is capable of suppressing the activation defect of Rad53 that occurs in the absence of Pkc1. In addition, we found that in HeLa cells Chk2 phosphorylation after genotoxic stress was affected when Pkcδ was specifically inhibited, suggesting that this isoform also plays a role in the DNA integrity checkpoint activation in mammalian cells. Historically, the most studied role of PKCδ has been the induction of apoptosis [46]. These studies have also generated evidence of a relationship between PKCδ and the response to DNA damage. It has been suggested its participation as an integrator of DNA damage signals upstream of the mitochondria [23]. Moreover, several connections between PKCδ and the DNA integrity checkpoint machinery are: PKCδ enters in the nucleus after DNA damage depending on ATM and phosphorylates the checkpoint protein RAD9 (member of 9-1-1 clamp complex) to activate apoptosis [47]; PKCδ also acts upstream of ATM and DNA-PKcs [48,49] and blocking its activity inhibits the phosphorylation of ATM and histone H2AX [48,50]; PKCδ overexpression induces S phase arrest and activation of the DNA integrity checkpoint [51]; and more recently, new PKCδ targets involved in the DNA damage response, repair, and activation of cell cycle checkpoints have been described in proteomic approaches [52]. The role of PKCδ in the regulation of cell survival or death has been discussed for years, but its role in cell cycle arrest has been clearly demonstrated in response to DNA damage in the early G1 phase through the induction of p53 and p21 [53,54,55], S phase [51], or G2/M phase [56]. Given all these functions, its complex relationship with cancer is evident [52,57] as it has generally been proposed as a tumor suppressor, yet oncogenic characteristics have also been attributed to it.

In this work, we intend to analyze the role of PKCδ in the regulation of the DNA integrity checkpoint in a non-tumoral but pluripotent cell model, mouse embryonic stem cells (mESCs). Then, we use the model yeast *S. cerevisiae* to identify other isoforms of the PKC family involved in the regulation of the DNA integrity checkpoint and to define their essential requirements to develop this function, taking advantage of the combination of both models to unravel molecular mechanisms involved in complex cellular processes.

## 2. Results

### 2.1. PKCδ Plays a Role in the DNA Integrity Checkpoint of mESCs

To date, PKCδ is the only described isoform of the mammalian PKC family capable of efficiently recovering genomic integrity checkpoint function in a yeast *pkc1* mutant [43]. In addition, the suppression of Pkcδ impairs the proper activation of this control mechanism in response to DNA damage in HeLa cells [43]. In this context, our aim was to analyze whether this role of PKCδ was also relevant in a mammalian embryonic stem cell model such as mESCs. These cells not only display distinct cycling dynamics and expression of cell cycle regulators [58] but must safeguard the integrity of their genetic information at all costs to ensure the correct differentiation of all the embryo cell types and prevent potential developmental defects or disorders. Seven clonal lines of mESCs were generated using CRISPR-Cas9 technology to eliminate PKCδ. Although PKCδ inactivation has been linked to inhibition of proliferation in some stem cell models [59], we did not detect changes in the basal growth or cell cycle distribution of the mutant ESCs lines compared to their controls (Appendix A). In order to characterize the role that PKCδ plays in the genomic integrity checkpoint of mESCs, we assessed by Western blot the phosphorylation of the checkpoint effector kinase CHK1 after acute exposure to the alkylating agent MMS. The cellular response was highly variable in both controls and mutant lines, but, notably, PKCδ-deficient mESCs showed a statistically significant reduction (*p*-value = 4.68 × 10^−5^) of 33% in the fraction of phosphorylated CHK1 compared to control cells, in data derived from a total of eight independent experiments including the seven clonal lines in different replicates (Figure 1). In view of these results, we conclude that PKCδ plays a role in the DNA integrity checkpoint of mESCs. Going further, the fact that the absence of PKCδ generates only a partial defect in the activation of CHK1 suggests that other PKC isoforms may be involved in this function. At this point, we decided to work on a yeast model to investigate this possibility.

### 2.2. PKCδ Partially Suppresses the Growth Defect of pkc1 Yeast Mutant Strains

The most well-known function of yeast Pkc1 is the maintenance of cellular integrity [60]. In addition, we have also described that Pkc1 is involved in the regulation of the DNA integrity checkpoint [43]. In fact, a temperature-sensitive *pkc1* mutant (*pkc1^ts^*) undergoes cell lysis at restrictive temperature and is deficient in activating the DNA integrity checkpoint effector kinase, Rad53. The mammalian protein PKCδ suppresses the defect in the checkpoint activation in the *pkc1^ts^
*mutant, but it is unable to recover the growth defect that the yeast strain presents at high temperatures [43]. More recently, in the course of our experiments, we analyzed whether PKCδ suppressed the growth defect in other *pkc1* mutants, specifically the *tetO_7_:PKC1* and *pkc1Δ* mutant strains. In the case of the *tetO_7_:PKC1* strain, in which *PKC1* is expressed under the control of the doxycycline-repressible *tetO_7_* promoter, PKCδ was able to partially rescue the growth defect caused by the absence of Pkc1 (Figure 2A). Interestingly, PKCδ was unable to suppress the growth defect of this mutant when temperature was increased at 37 °C. This is consistent with our previous observation that PKCδ failed to restore growth of the *pkc1^ts^
*mutant. In the case of a *pkc1Δ* mutant, PKCδ was also capable of suppressing its growth defect in a medium without sorbitol (Figure 2B). In conclusion, these results indicate that PKCδ is indeed able to suppress not only the defect in the activation of the DNA integrity checkpoint but also the growth defect of different *pkc1* mutants, although this second capacity is lost at high temperatures, probably because the cell lysis defect in the absence of Pkc1 is more severe when temperature increases. So, we had two functional tests useful to identify the mammalian PKC isoforms capable of performing yeast Pkc1 functions.

### 2.3. The Novel Isoforms PKCδ and PKCθ Are the Only Ones Capable of Performing Pkc1 Functions but with Different Efficiency

We previously studied the functionality in yeast cells of at least one member of the three groups of the mammalian PKC family [43]. Here, we have completed this functional analysis including all the isoforms (Figure 3A). Specifically, we evaluated the ability of the PKC isoforms to suppress the defects in DNA integrity checkpoint activation and growth of *pkc1* mutant strains. First, the *pkc1^ts^* strain was transformed with plasmids expressing the different isoforms, and we analyzed by Western blot the phosphorylation pattern of the checkpoint effector kinase Rad53 in response to the genotoxic agent MMS. Our results showed that only the expression of PKCδ and, interestingly, of its phylogenetically closest isoform PKCθ, allowed the activation of Rad53 after the induction of genotoxic stress, although to a lesser extent (Figure 3B). Second, the suppression of the growth defect of the mutant strain *tetO_7_:PKC1* by the different isoforms of PKC was evaluated. Again, only the expression of PKCδ and, to a lesser extent PKCθ, rescued the lethality of *pkc1* mutant cells (Figure 3C). Therefore, we can conclude that PKCδ and PKCθ are the only mammalian isoforms capable of replacing yeast Pkc1 in the activation of the DNA integrity checkpoint and growth control, although with differences in their functionality, PKCθ being less efficient than PKCδ.

We then asked if the different functionality observed between PKCδ and PKCθ in yeast could be explained by differences in their protein levels; however, no significant differences were observed, neither under normal conditions nor in the presence of DNA damage (Appendix A). This implies that intrinsic differences in functionality should exist. Next, we decided to characterize the molecular basis of these differences in our yeast model.

### 2.4. PKCδ and PKCθ Have Different Requirements of Activation Loop Phosphorylation for DNA Integrity Checkpoint Regulation

PKCs typically require phosphorylation of a Thr residue in their activation loop to be catalytically competent [18]. However, it has been described that PKCδ is a particular case since this phosphorylation (in Thr^505^ in the case of PKCδ) is not essential for its catalytic activity due to the nearby presence of an acidic residue (Glu^500^) [61,62]. Interestingly, PKCθ also contains an acidic residue (Asp^533^) close to the Thr^538^ of the activation loop; nevertheless, it has been reported that PKCθ activation depends on phosphorylation of the Thr^538^ residue [63].

We asked whether PKCδ and PKCθ regulation of the DNA integrity checkpoint and growth in yeast requires the phosphorylation of the activation loop. By site-directed mutagenesis, Thr^505^ and/or Glu^500^ of PKCδ were replaced by Ala (mimicking a non-phosphorylatable state) and Gly (amino acid present in this same position in the non-functional novel PKCε isoform), respectively. In the absence of Thr ^505^, phosphorylation of checkpoint kinase Rad53 was still observed, but to a lesser extent. Therefore, the phosphorylation in Thr^505^ of PKCδ is not essential for the activation of the DNA integrity checkpoint, although it is necessary for an efficient activation. On the other hand, Glu^500^ does not contribute to PKCδ functionality since the substitution of this distal acidic residue by Gly had no effect on the ability of PKCδ to activate Rad53 in response to DNA damage, neither in the wild-type protein nor in the PKCδ^T505A^ mutant protein (Figure 4A, left panel). A similar result was obtained in growth assays in which the expression of both single and double mutants was able to recover the growth defect of a *pkc1* mutant strain (Figure 4A, right panel).

In the case of PKCθ, cells expressing the mutant protein PKCθ^T538A^, in which the Thr^538^ residue was replaced by Ala, were not able to activate Rad53 in response to MMS (Figure 4B, left panel). Thus, phosphorylation of the PKCθ activation loop is necessary for its function in the response to genotoxic stress. Similarly, growth assays revealed that this phosphorylation was also necessary to recover the viability of a *pkc1* mutant strain (Figure 4B, right panel).

In summary, our results indicate that activation loop phosphorylation is essential for PKCθ functionality in yeast cells, whereas it is not necessary in the case of PKCδ, although it may play a role in the optimal response to DNA damage.

### 2.5. Tyr Residues of PKCδ Phosphorylated in Oxidative Stress Are Not Essential for Activating the DNA Integrity Checkpoint

PKCδ is, of all mammalian PKCs, the most phosphorylated isoform in Tyr residues. It has been described that in response to H_2_O_2_, PKCδ is phosphorylated in Tyr^311^, Tyr^332^, and Tyr^512^, increasing its protein activity in both in vitro and in vivo assays [64,65]. The treatment with H_2_O_2_ generates oxidative stress which, in turn, may cause DNA damage, so these Tyr phosphorylations could be associated with genotoxic stress. In order to identify whether these three Tyr residues are relevant to PKCδ function in the DNA integrity checkpoint, they were replaced by site-directed mutagenesis to Phe (mimicking a non-phosphorylatable state) both in the wild-type protein and the PKCδ^T505A^ mutant protein. When we analyzed their ability to activate Rad53 in response to DNA damage, no differences were detected in any case (Appendix A). This indicates that these Tyr residues are not involved in the function of PKCδ in the response to DNA damage.

### 2.6. The Catalytic Fragment of PKCδ, but Not That of PKCθ, Is Sufficient for the Activation of the DNA Integrity Checkpoint in Yeast

In a parallel approach, we tried to delimit the regions of the protein required for the functionality of the PKCδ and PKCθ isoforms. We intended to identify the minimal functional versions by cloning truncated proteins consisting only in the catalytic domain: PKCδ^CF^-GFP and PKCθ^CF^-GFP. The expression of PKCδ^CF^, but not PKCθ^CF^, allowed Rad53 phosphorylation in a *pkc1*^ts^ mutant in response to DNA damage (Figure 5A, top panel). Thus, we can conclude that the catalytic fragment of PKCδ is sufficient to activate the DNA integrity checkpoint. Similarly, only the expression of PKCδ^CF^ was able to partially suppress the growth defect of the *pkc1* mutant strain (Figure 5A, left bottom panel). These differences in functionality between both catalytic fragments are not due to differences in their protein levels (Figure 5A, right bottom panel), so must be due to intrinsic differences in their activity.

Next, we considered whether the differences observed between PKCδ and PKCθ catalytic fragments could have a structural basis. The tertiary structure of the catalytic fragment of human PKCθ in the presence of competitive inhibitors is known [66]; however, to date, no structure of the catalytic fragment of PKCδ is available. To obtain predictions of the structure of both catalytic fragments, we used the protein structure modeling server *I-TASSER*. Modeling of the catalytic fragments of PKCδ and PKCθ generated very similar structures corresponding to conventional structure of the catalytic domain of kinases with a central cleft corresponding to the active center (Figure 5B). Therefore, there are no apparent structural differences that would explain the important differences in functionality.

In this context, we decided to study the importance of phosphorylation of the activation loop but now in the minimal functional version of PKCδ. Thr^505^ and Glu^500^ were replaced by Ala (PKCδ^CF T505A^) or/and Gly (PKCδ^CF E500G^, PKCδ^CF T505A,E500G^), respectively, in the construction containing the catalytic fragment of PKCδ. The results showed that mutation of Thr^505^ impaired activation of the DNA integrity checkpoint, whereas mutation of Glu^500^ had a minor role in Rad53 phosphorylation (Figure 5C, top panel). The same results were observed in cell growth assays: while *pkc1* mutant cells expressing the wild-type version of PKCδ^CF^ or the PKCδ^CF E500G^ mutant showed similar growth, mutation of Thr^505^ was deleterious for growth, although it did not completely eliminate the ability of PKCδ^CF^ to alleviate the growth defect of a *pkc1* mutant strain (Figure 5C, bottom panel). This loss of functionality was not due to changes in protein levels (Appendix A). In conclusion, all these observations indicate that phosphorylation of Thr^505^ is essential for PKCδ^CF^ functionality to activate the DNA integrity checkpoint and to sustain cell growth.

Given the previous result, we envisage the possibility that the lack of functionality observed for the catalytic fragment of PKCθ could be due to a failure in the phosphorylation of its activation loop. To test this, Thr^538^ of PKCθ was replaced by Glu (PKCθ^CF T538E^) to mimic a phosphorylated state. As expected, protein levels were not affected by changes in the phosphorylation status of the activation loop (Appendix A). However, the expression of this version of PKCθ^CF^ was not able to recover the DNA integrity checkpoint activity (Figure 5D, top panel). Consistently, the phosphomimic version of PKCθ^CF^ did not improve cell growth (Figure 5D, bottom panel). These results suggest that the lack of function of PKCθ^CF^ compared to PKCδ^CF^ is not caused by defects in its phosphorylation in the activation loop.

### 2.7. The A-Helix of PKCδ Plays a Role in Activating the DNA Integrity Checkpoint by Providing an Alternative Mechanism to Activation Loop Phosphorylation

As mentioned above, PKCδ is a particular member of the PKC family in terms of its activation mechanisms. PKCδ presents an alternative and exclusive activation system involving the so-called A-helix [67]. In order to study the role of A-helix in the response to DNA damage, we added to the N-terminal of the catalytic fragment of PKCδ an extension of 35 amino acids that includes the A-helix (PKCδ^Ahelix-CF^-GFP). The ability of this truncated version to suppress the defects in the activation of the DNA integrity checkpoint and growth of *pkc1* mutant cells was investigated. The results showed that the PKCδ^Ahelix-CF^-GFP construction was functional since its expression in yeast cells allowed Rad53 to be correctly activated in response to genotoxic stress and to suppress the cell growth defect of the *pkc1* mutant strain in the same way as PKCδ^CF^ (Figure 6A).

Next, we studied the need for phosphorylation of the activation loop when the A-helix was present. Thr^505^ and Glu^500^ residues in PKCδ^Ahelix-CF^ were replaced by Ala and Gly, respectively, and functional analyses of single and double mutants were carried out. Both PKCδ ^Ahelix-CF T505A^ and PKCδ^Ahelix-CF T505A,E500G^ were able to efficiently activate the DNA damage response and also to support cell growth in *pkc1* mutants (Figure 6B). Thus, the phosphorylation of the activation loop is no longer necessary if the alternative activation mechanism involving A-helix is present.

It was described that the mechanism in which the A-helix participates allows the stabilization of the activation loop by a series of interactions between specific residues, located in A-helix, C-helix, activation loop, and F-helix. They are: (1) Tyr^332–^Phe^498^, (2) Trp^336–^Arg ^397^, and (3) Ile^497^–Phe^525^ [67]. The structure of PKCδ^Ahelix-CF^ was modeled and compared with the PKCδ^CF^. The analysis revealed that the presence of the A-helix causes significant conformational changes in the region of the activation loop (Figure 7A). Specifically, residues Ile^497^ and Phe^525^ flanking the Thr^505^ of the activation loop, which are spatially distant in the absence of the A-helix, are reoriented to a close proximity when the A-helix is added. Additionally, the side chain of Phe^498^ (also located near the activation loop) undergoes a reorientation.

Some of the indicated amino acids involved in the A-helix mechanism are only present in PKCδ, which could explain that this mechanism is exclusive for this PKC isoform. To study this possibility, Tyr ^332^, Trp^336^, Ile^497^, Phe ^498^, and Phe^525^ were replaced in PKCδ^Ahelix-CF T505A,E500G^ (so that the functionality of the protein depended solely on the mechanism involving the A-helix) by those present in the same position in PKCθ or, in the case of conserved residues, by those present in PKCε, the phylogenetically closest isoform to PKCδ unable to activate the DNA integrity checkpoint (Appendix A). Unexpectedly, the mutant protein obtained (PKCδ^Ahelix-CF T505A,E500G,Y332Q,W336R,I497M,F498L,F525H^) still conserved the functionality to properly activate Rad53 in response to MMS in a *pkc1^ts^* mutant strain (Figure 7B). This indicates that the activation mechanism driven by A-helix that allows the activation of the DNA integrity checkpoint does not strictly depend on the amino acid present in PKCδ since the substitution by other amino acids does not abrogate this ability.

It has to be noted that the suggested structural changes associated with the addition of the A-helix commented on above involve the reorientation of two residues of Phe and a Ile, all of them amino acids with a bulky side chain. This led us to consider that the size of side chains of key amino acids close to Thr^505^ could be decisive to stabilize the activation loop allowing PKCδ activation in the absence of Thr^505^ phosphorylation. Interestingly, the residues of Met, Leu, and His of PKCθ that replace in our construction Ile^497^, Phe^498^, and Phe^525^, respectively, also have bulky side chains, which could explain why the functionality of the protein is conserved. According to this hypothesis, the presence of amino acids with small side chains should affect the functionality of PKCδ^Ahelix-CF^. The structural modeling showed that the replacement of the residues Ile^497^, Phe^498^, and Phe^525^ by Ala rendered a less compacted conformation around the activation loop that could affect its stabilization (Figure 7A). To test the hypothesis, Ile^497^, Phe^498^, and Phe^525^ were mutated to Ala in PKCδ^Ahelix-CF T505A^. Importantly, the analysis of Rad53 phosphorylation in response to MMS revealed that replacing these bulky residues with an amino acid with a small side chain dramatically reduced the ability of PKCδ^Ahelix-CF T505A^ to activate the DNA integrity checkpoint. This loss of functionality is not due to a lower amount of protein since there were no significant differences in the protein levels of both constructions (Figure 7C).

Overall, our results indicate that the presence of the A-helix in PKCδ causes a reorientation in bulky residues close to the activation loop that could allow its stabilization. This mechanism is sufficient to maintain the functionality of PKCδ in the absence of phosphorylation of the Thr^505^ residue in yeast cells in order to activate the DNA integrity checkpoint.

Finally, as the substitution of PKCδ-specific residues with those of PKCθ did not affect the functionality of the protein, we considered the possibility that the addition of the A-helix would confer to the catalytic fragment of PKCθ the ability to activate the DNA integrity checkpoint. However, this was not the case since the PKCθ^δAhelix-CF^ construction was unable to activate Rad53 (Figure 8A). PKCθ^δAhelix-CF^-GFP protein levels did not justify the failure in the functionality of the construction (Figure 8B and Appendix A). We wonder if the different functionality between PKCδ^Ahelix-CF^ and PKCθ^δAhelix-CF^ could be caused by differences in their tertiary structure. Interestingly, the predicted tertiary structures revealed a critical difference: in the case of PKCθ^δAhelix-CF^, the A-helix sequence remains as an unstructured flexible region, whereas in the case of PKCδ^Ahelix-CF^, it folds as the expected α-helix structure (Figure 8C). This result suggests that other regions of the protein are involved in the A-helix folding and that the correct folding of the A-helix would be essential for the stabilization effect over the activation loop to activate the kinase.

## 3. Discussion

Different model organisms have been used to deepen the understanding of cell proliferation. Many reasons have made the budding yeast *Saccharomyces cerevisiae* a key model for understanding and characterizing the cell cycle [68]. Cell cycle regulation mechanisms are highly conserved among eukaryotes. Because of that, it is not surprising that studies in yeast have elucidated fundamental mechanisms in cell cycle control that are later extended to higher eukaryotes. The DNA integrity checkpoint is not an exception and, thus, it has been described that the main regulators that participate in these signaling pathways play similar roles across species, from humans to yeasts [12].

In this work, the utilization of yeast enabled us to further analyze which of the nine isoforms of the mammalian PKC family participate in the regulation of the DNA integrity checkpoint. In the case of multiprotein families (like the PKC family), an important advantage of heterologous expression systems is that they allow performing functional studies of individual isoforms in the absence of the others. The *S. cerevisiae pkc1* mutant provides a valuable opportunity to conduct different functional assays of mammalian PKCs. In this work, we have developed two functional assays: one for cell viability and the other for DNA integrity checkpoint activation by monitoring the phosphorylation of the yeast effector kinase Rad53. Through these experimental approaches, we have identified that PKCδ and PKCθ are the only isoforms capable of activating the checkpoint in yeast when Pkc1 is absent. Functional redundancy is a notable characteristic of cell cycle regulation [69]. The cell must ensure that in the absence of one regulator of an essential process, another protein can perform its function, thereby enhancing the robustness of the control mechanisms. It is important to note that functional redundancy does not necessarily imply identical efficiency or activation mechanisms. In fact, we have described that PKCδ is more efficient than PKCθ in this function.

PKCδ and PKCθ can undergo two types of modifications that contribute to their regulation: reversible modifications, which include phosphorylation at Ser/Thr residues and Tyr residues, and irreversible modifications, such as caspase-mediated proteolytic cleavage. In addition, the existence of a mechanism exclusive to PKCδ involved in its activation through the stabilization of its activation loop has been described [25].

A crucial phosphorylation event occurs at a Thr residue on the activation loop catalyzed by PDK-1. This event stabilizes the activation loop and ensures proper positioning of residues responsible for catalysis. In the context of the complete protein, we have observed that functionality in yeast of PKCθ requires the phosphorylation of the Thr in the activation loop; in contrast, this phosphorylation is dispensable in the case of PKCδ, although it contributes to optimal function. The fact that the activity is reduced when the Thr in the activation loop is mutated indirectly suggests that the Pkh1 and Pkh2 kinases of *S. cerevisiae*, which are homologous to the PDK-1 kinase in mammalian cells [18], are able to phosphorylate and activate mammalian PKCs. Besides, the fact that PKCδ is still active when Thr in the activation loop is mutated points to the existence of additional mechanisms exclusive to PKCδ. In a first approach, our results have allowed us to discard the possibility that this alternative mechanism involves the phosphorylation in Tyr residues described in response to oxidative stress.

In search of essential domains involved in the activation of the checkpoint, we obtained truncated versions of PKCδ and PKCθ consisting exclusively of their catalytic domain. It is known that in response to apoptotic signals, such as high-intensity ionizing radiation and DNA-damaging agents, caspase-3 carries out a proteolytic cleavage of PKCδ at a specific site located in the hinge region (GSDILD^327^N), generating a 40 kDa catalytic fragment [49,70,71]. As a result of this processing, a constitutively active PKCδ catalytic domain is released, which mediates the induction of apoptosis [72]. We have observed that the PKCδ catalytic fragment is sufficient to activate the DNA integrity checkpoint when expressed in yeast. This demonstrates that this truncated version of the protein is functional and suggests that its processing by caspases under specific conditions would allow the existence of a functional version that fulfills its role in the DNA integrity checkpoint control. As regards PKCθ, a proteolytic cut by caspase-3 has also been described in response to apoptotic stimuli in humans. However, there is controversy about the function of this processing. Some authors propose that, similar to PKCδ, the cleavage leads to the activation of PKCθ [73,74], whereas others state that it causes the degradation of the PKCθ catalytic fragment [75]. It is necessary to comment that the caspase cleavage site in human PKCθ (DEVD^354^K) is not conserved in *Mus musculus* PKCθ (the one used in this study), where Asp^354^ residue is replaced by Asn, which could indicate differences in the PKCθ regulation between human and mouse. Unlike PKCδ, we have observed that the PKCθ catalytic fragment is not able to activate the checkpoint when expressed in yeast. This supports the idea that this truncated version of the protein is not functional.

Revisiting the need for phosphorylation of the activation loop, in this minimal context we found that the functionality of the catalytic fragment of PKCδ depends mainly on the phosphorylation of Thr^505^. It can be envisaged that the lack of functionality of PKCθ^CF^ could be caused by the inability of yeast Pkh1,2 to phosphorylate Thr^538^ in the PKCθ^CF^ activation loop. Nevertheless, the fact that the expression of a phosphomimic version of PKCθ^CF^ did not alleviate the defects, neither in the activation of the DNA integrity checkpoint nor growth of *pkc1* mutant cells, suggests that this would not be the case. We also discarded that protein levels explain these differences because the non-functional PKCθ^CF^ protein presents the same or even higher levels than PKCδ^CF^. Rather, we propose that PKCδ^CF^ can recognize and phosphorylate specific DNA integrity checkpoint substrates that PKCθ^CF^ cannot. This could reflect intrinsic differences between the catalytic fragments in order to act on potential substrates. Alternatively, this could be influenced by differences in the subcellular localization. The subcellular localization of mammalian PKCs varies according to the stimuli received by the cell. It has been reported that both PKCδ and PKCθ are capable of localizing to the nucleus, and the signals responsible for this nuclear accumulation have been described. Interestingly, in the case of PKCδ, proteolytic cleavage by caspase-3 exposes a bipartite NLS located in the C-terminal tail of the catalytic fragment [76,77,78]; however, in the case of PKCθ, nuclear entry of the complete protein depends on a motif located in the C1b regulatory domain [79], which is absent in the truncated catalytic fragment version. Future comparative interactomic studies and subcellular localization analysis between PKCδ^CF^ and PKCθ^CF^ will address these aspects.

As mentioned above, PKCδ presents a specific mechanism implied in the DNA integrity checkpoint activation. Our study has identified that the presence of the A-helix renders the phosphorylation of the activation loop dispensable for PKCδ functionality. It has to be noted that the replacement of amino acids involved in this mechanism by those residues present in PKCθ does not affect the functionality of PKCδ^Ahelix-CF^. It was previously described that the replacement of the two Phe by the residues present in PKCθ in human PKCδ^Ahelix-CF T505A^ reduces the kinase activity of the protein [67]. Our results do not rule out that there is a variation in the kinase activity of the PKCδ ^Ahelix-CF T505A, E500G, Y332Q, W336R, I497M, F498L, F525H^ mutant, but if this is indeed the case, the fact that we do not observe differences in the phosphorylation of Rad53 revealed that the remaining kinase activity is sufficient to correctly activate the response to DNA damage. Interestingly, the results obtained in this work indicate that the replacement of three of these residues (Ile^497^, Phe^498^, and Phe^525^) for smaller amino acid does affect the functionality of PKCδ^Ahelix-CF^, resulting in the loss of its ability to activate the DNA integrity checkpoint in *S. cerevisiae* cells. This result highlights the importance of the side chain size of the residues nearby the activation loop to ensure the functionality of the protein. This could explain why the bulky residues of PKCθ are able to replace PKCδ ones in key positions for the A-helix mechanism. However, it has to be noted that the catalytic fragment of PKCθ is still not functional upon the addition of the PKCδ A-helix. We have discarded that this was due to reduced protein levels. Remarkably, the correct folding of this region as an α-helix does not occur in the context of the PKCθ isoform as deduced from the modeling analysis, which could explain the lack of functionality. Additionally, differences in substrate specificity or subcellular localization could be considered, as mentioned above. In this line, the contribution of the PKCδ A-helix to the catalytic activity, the recognition of substrates, or the localization of the protein are possibilities that will be addressed in future studies.

Finally, one central conclusion of our work is that our pluripotent mESCs model evidences the involvement of PKCδ in the DNA integrity checkpoint control. Moreover, in light of our conclusions in yeast revealing PKCδ and PKCθ isoforms as the only players in the control of the checkpoint, the fact that inactivation of PKCδ in mESCs does not eliminate completely the checkpoint function allows us to propose that PKCθ would be responsible for the remaining activity. Returning to the actual context of the mammalian PKC isoforms, it is important to note that they present different tissue distribution. While PKCδ is ubiquitously expressed in mammalian tissues [80,81], the expression of PKCθ is restricted to a few cell types, including T lymphocytes, platelets, and skeletal muscle cells [82,83]. As mentioned in the introduction section, PKCδ has been widely related to regulation of cell survival or death and diseases connected with these processes. Regarding PKCθ, its best characterized function is in T-cell activation and survival [84]. In addition, PKCθ participates in the reorganization of the cytoskeleton in the immune synapse, as well as associates with chromatin to control the expression of genes involved in the regulation of cytokines [79,85]. Beyond its role in the immune response, there is increasing evidence linking PKCθ with the development of different diseases, especially autoimmune disorders and various types of cancer [86]. Stem cells from adult tissues persist throughout life, and therefore their response to DNA damage is essential to protect their role in the tissue cell hierarchy. In addition, the tumorigenic transformation of these cells is at the basis of many types of tumors and leads to an alteration in the mechanisms of response to damage that facilitates tumor progression [87]. Therefore, future work would entail exploring the relevance of PKCθ in the DNA integrity checkpoint activation not only in mESCs but also in cell models such as somatic stem cells from specific tissues. Of particular interest is the case of adult neural stem cells in which we have obtained preliminary data showing high expression of PKCθ. It has been postulated that gliomas, the most common malignant brain tumors, highly invasive and resistant to conventional therapy, may be the result of malignant transformation of neural stem cells [88,89]. Interestingly, other PKC isoforms have been proposed as targets in glioblastoma treatment [90]. To elucidate the role of PKCθ in neural stem cells could open new therapeutic strategies.

## 4. Materials and Methods

### 4.1. Mouse Embryonic Stem Cells Transfection and Growth Conditions

mESCs derived from the E14Tg2a cell line [91] were grown on 0.1% gelatine-coated plates in culture medium containing GMEM (Sigma-Aldrich, Burlington, Massachusetts, United States. Cat#G5154), 10% fetal bovine serum (Capricorn Scientific, Ebsdorfergrund, Germany, Cat#FBS-12A), 100 units/mL of penicillin-streptomycin (Invitrogen, Waltham, MA, USA, Cat#15140148), 2 mM of L-glutamine (Gibco, Paisley, Scotland, Cat#25030-024), 1 mM of Sodium Pyruvate (Biowest, Nuaillé, France, Cat#L0642-500), and 1× non-essential amino acids (NEAA) (Gibco, Paisley, Scotland, Cat#11140035) supplemented with 100 U/mL of leukemia inhibitory factor (LIF) (homemade) and 0.1 mM of β-mercaptoethanol ( Sigma-Aldrich, Burlington, MA, USA, Cat#M6250). These cells were maintained at 37 °C in a 5% CO_2_ humidified incubator and were routinely passaged every 2 or 3 days, when 70–80% confluence was reached.

mESCs *Pkcδ^−/−^* refers to clonal lines that were generated using CRISPR/Cas9 technology by nucleofection with the pCas9_GFP plasmid (Addgene, Watertown, MA, USA, Cat#44719) along with a guide plasmid which was constructed based on the pU6-gRNA plasmid containing a commercial guide targeting exon 2 of the *Pkrcd* gene (Target Site: TTGAAGGAGATGCGCAGGAAGG; SIGMA CRISPR Plasmid, Sigma-Aldrich, Burlington, MA, USA). As a control, *Pkcδ^+/+^* cells were nucleofected only with the pCas9_GFP plasmid. The Mouse Neural Stem Cell Nucleofector^TM^ Kit (Lonza, Basel, Switzerland, Cat#VPG-1004) in combination with the Nucleofector 2b device (Amaxa, Lonza, Basel, Switzerland) were employed for this purpose. GFP^+^ cells were purified by FACS with a FACSAria Fusion cell sorter (BD, Franklin Lakes, NJ, USA), and single-cell seeding was used to initiate independent clonal lines. Once expanded, the clones were analyzed by PCR and Sanger sequencing to identify specific mutations and by RT-qPCR with specific primers and Western blot for analyzing PKCδ expression.

### 4.2. Yeast Strains and Growth Conditions

The yeast strains used in this study are three different *pkc1* mutant strains: the *pkc1* thermosensitive mutant (*pkc1^ts^*) JC6-3a (*MATa pkc1-8 ade2-1 trp1-1 leu2-3,112 his3-11,15 ura3-52 can1 met^-^*), the deletion mutant (*pkc1Δ*) GPY1115b (*MATa PKC1::HIS3 leu2-3,112 ura3-52 his3-Δ200 trp1-Δ901 ade2101 suc2-Δ9*) kindly provided by Dr. G. Paravicini, and the conditional *pkc1* mutant in which the *PKC1* promoter has been substituted by the *tetO_7_* promoter (*tetO_7_:PKC1*) JCY1471 (*tTR’::LEU2 tetO7::PKC1-kanMX4* inserted in W303-1a), all of them used previously [43]. W303-1a (*MATa ade2-1 trp1-1 leu2-3,112 his3-11,15 ura3-52 can1*) was used for plasmid construction.

Cells were grown on standard yeast extract–peptone–dextrose (YPD) or YPD medium supplemented with 1 M of sorbitol where indicated. For induction of genotoxic stress in liquid cultures, 0.04% MMS was added to exponentially growing cultures for 1 h. To repress the *tetO_7_* promoter, doxycycline was added to a final concentration of 10 μg/mL.

### 4.3. Yeast Plasmids

Centromeric plasmids pPKC1, pPKCα, pPKCε, pPKCδ, pPKCη, pPKCι, and pPRK2 plasmids were constructed as previously described [43]. The rest of the isoforms were cloned into a PKC1p plasmid that contained the PKC1 promoter and the ADH1 terminator, as described below.

The PKC1_p_ plasmid was constructed in a two-step process. First, the *ADH1* terminator including the stop codon (+772, +1011), amplified from pFA6a-GFP(S65T) with oligos containing a SalI or PstI site, was cloned in SalI-PstI-digested YCplac33. Next, *PKC1* promoter (−600, −1) amplified from genomic DNA with oligos containing the EcoRI and KpnI restriction sites was introduced by EcoRI-KpnI digestion.

The pPKCθ, pPKCβ, pPKCγ, pPKCζI, and pPKCζII plasmids were constructed by cloning in KpnI-SalI-cleaved pPKC1_p_ a DNA fragment corresponding to the coding region of the referred isoform obtained by PCR amplification from mouse cDNA using a forward oligo containing a KpnI restriction site and a reverse oligo containing a SalI site. The complete plasmid collection expressing the PKC family mouse isoforms is currently available in the non-profit plasmid repository (https://www.addgene.org/Juan_Carlos_Igual/ (accessed on 27 September 2023).

The pPKCδ-GFP plasmid was obtained by integrating the PKC1 promoter and the pPKCδ coding region in pGFP-ADH1t plasmid. The plasmid pPKCθ-GFP was obtained by recombination [92] in the W303-1a wild-type *S. cerevisiae* strain cotransforming with the plasmid pPKCδ-GFP linearized with KpnI/BamHI and the PKCθ coding sequence, obtained by PCR from the pPKCθ plasmid. The plasmids pPKCδ^CF^-GFP and pPKCθ^CF^-GFP, which express the catalytic fragments of PKCδ (amino acids 346–674) and PKCθ (amino acids 378–707), were also obtained by recombination in the W303-1a strain cotransforming with the plasmid pPKCδ-GFP linearized with KpnI/BamHI and the catalytic fragments obtained by PCR from pPKCδ and pPKCθ plasmids.

The plasmid pPKCδ^Ahelix-CF^-GFP that express the catalytic fragment of PKCδ with the N-terminal extension of 35 amino acids that includes A-helix (amino acids 311–345) was also obtained by recombination in the W303-1a strain cotransforming with the plasmid pPKCδ-GFP linearized with KpnI/BamHI and the PKCδ fragment obtained by PCR from pPKCδ plasmid. The plasmid pPKCθ^Ahelix-CF^-GFP that expresses the catalytic fragment of PKCθ with the N-terminal extension of 35 amino acids that includes the pPKCδ A-helix (amino acids 311–345) was also obtained by recombination in the W303-1a strain cotransforming with the plasmid pPKCθ-GFP linearized with KpnI/BamHI and the PKCδ fragment obtained by PCR from pPKCδ plasmid.

All mutant versions of these plasmids were obtained by site-directed mutagenesis of the indicated protein (Appendix A) using the QuikChange^®^ Site-directed Mutagenesis Kit (Agilent Technologies, Santa Clara, CA, USA) with the oligonucleotides listed in Appendix A.

All constructed plasmids were checked by sequencing, and at least 2 independent clones of each construction were analyzed in each experiment (a representative experiment is shown).

### 4.4. Western Blot Analysis

Yeast total protein extracts were prepared as previously described [45]. For mESCs total protein extracts, cells were collected in 0.1% PBS, after centrifugation, 100 μL of sample buffer (0.2 M Tris-HCl pH 7.5, 8% SDS, 40% glycerol, 0.2 M DTT, 0.04% bromophenol blue) was added and samples were incubated for 5 min at 95 °C. Equivalent amounts of protein were resolved in an SDS–PAGE gel and transferred onto a nitrocellulose membrane. The primary antibodies used in this study include rabbit polyclonal anti-Rad53 (Abcam, Waltham, Boston, FL, USA; Cat#ab104232), mouse mAb anti-GFP (Roche, Vienna, Austria, Cat#11814460001), rabbit polyclonal anti-PKCδ (Cell Signaling, Danvers, MA, USA, Cat#2058), rabbit mAb anti-phospho-CHK1 (Ser345) (Cell Signaling, Danvers, MA, USA, Cat#2348), and mouse mAb anti-Chk1 (Cell Signaling, Danvers, MA, USA, Cat#2360). Blots were developed with HRP-labeled secondary antibodies using the Supersignal^TM^ West Femto Maximum Sensitivity Substrate (Thermo Fisher Scientific, Waltham, MA, USA), and bands were detected and quantified with an ImageQuant^TM^ LAS 4000mini Biomolecular Imager (GE Healthcare, Chicago, IL, USA).

## 5. Conclusions

This multimodel study has allowed us to demonstrate the involvement of PKCδ in the control of the DNA integrity checkpoint in non-tumor mammalian cells, mESCs, and has opened the possibility to analyze whether other isoforms of the PKC family could also be involved. The *S. cerevisiae* yeast model has allowed us to demonstrate that PKCθ is also involved in the regulation of the response to DNA damage and to study the differences in efficiency and the molecular requirements for this function between PKCδ and PKCθ. We describe that PKCδ is more efficient in activating the checkpoint, probably due to the presence and correct folding of the A-helix, which provides an alternative activation mechanism independent of the phosphorylation of the activation loop. The materials and strategies developed allow us to design future studies to further analyze the role of PKC in the DNA integrity checkpoint in both yeast and mammals.

## Figures and Tables

**Figure 1 ijms-24-15796-f001:**
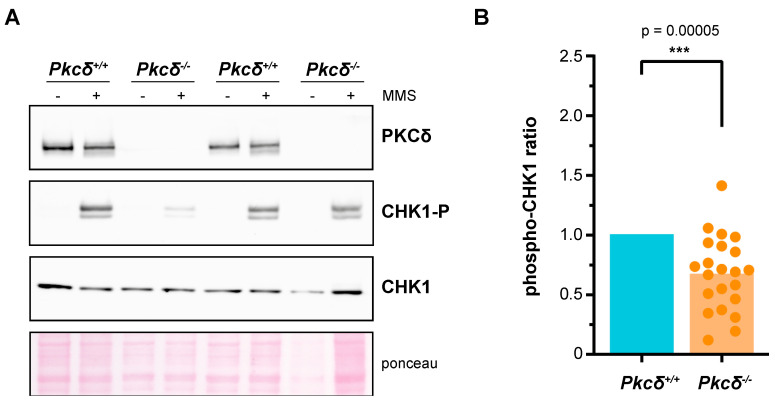
Analysis of the role of PKCδ in the activation of the DNA integrity checkpoint of murine ESCs. In each experimental replicate, different clones of both *Pkcδ*^−/−^ cells and *Pkcδ*^+/+^ cells were independently expanded in culture over two or three passages. The same number of cells were seeded in all cases, and, two days later, they were treated with DMSO or MMS (0.04%, 2 h). Protein extracts were obtained, and the presence of PKCδ as well as the levels of CHK1 and phosphorylated CHK1 in Ser^345^ were analyzed by Western blot. The phospho–CHK1 ratio (phosphorylated CHK1 relative to total CHK1) was quantified. Each experimental replicate was internally normalized to the *Pkcδ^+/+^* samples. A total of 16 control samples and 22 *Pkcδ*^−/−^ samples were analyzed, corresponding to 3 independent clones of *Pkcδ^+/+^* and 7 independent clones of *Pkcδ*^−/−^ cells. A representative Western blot is shown in (**A**), and all the replicates are represented in (**B**).

**Figure 2 ijms-24-15796-f002:**
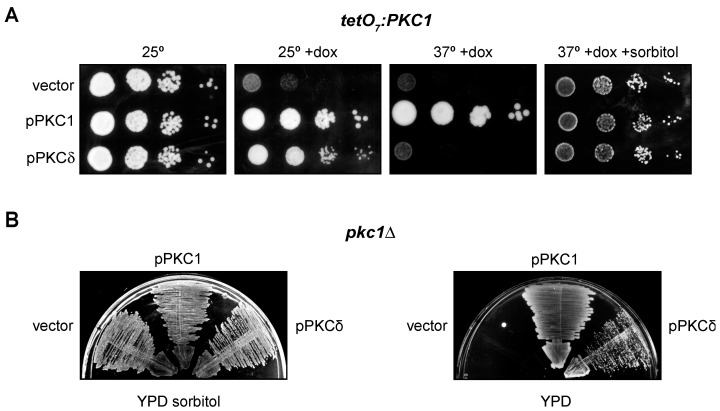
Suppression of the growth defect of *pkc1* mutants by PKCδ. (**A**) Serial dilutions of exponentially growing cultures of the *tetO_7_:PKC1* (JCY1471) strain transformed with plasmids pPKC1, pPKCδ or an empty vector were spotted onto YPD plates supplemented with doxycycline 10 μg/mL and sorbitol 1 M when indicated and incubated at 25 °C or 37 °C for 3 days. (**B**) Cells of the *pkc1∆* (GPY1115) strain transformed with plasmids pPKC1, pPKCδ or an empty vector were grown on YPD plates or YPD plates supplemented with sorbitol 1 M and incubated at 25 °C for 3 days.

**Figure 3 ijms-24-15796-f003:**
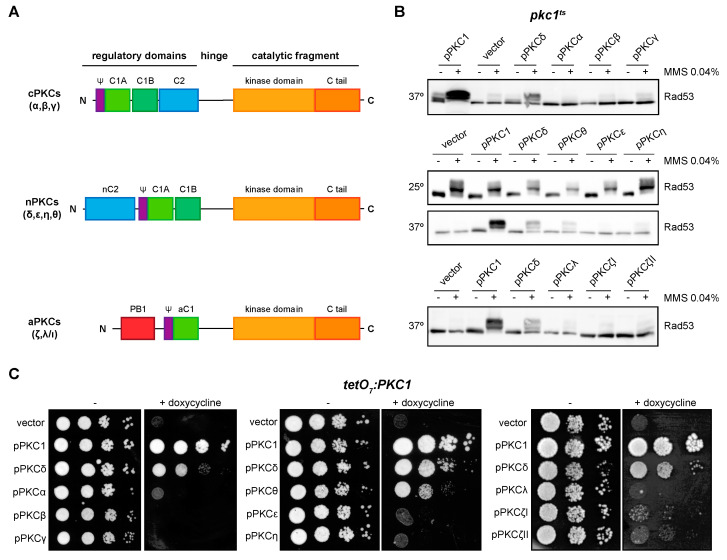
Suppression of DNA integrity checkpoint activation and growth defects of *pkc1* mutant strains by mammalian PKCs. (**A**) Schematic representation of classical (α, β, γ), novel (δ, ε, η, θ), and atypical (ζ, λ/ι) PKC proteins. (**B**) Exponentially growing cultures of the *pkc1^ts^* (JC6-3a) strain transformed with plasmids pPKC1, pPKCα, pPKCβ, pPKCγ, pPKCδ, pPKCθ, pPKCε, pPKCη, pPKCλ, pPKCζI, pPKCζII or an empty vector were transferred to 37 °C for 3 h and then incubated in the absence or presence of MMS 0.04% for 1 h. Cultures of the *pkc1^ts^* strain transformed with plasmids pPKCδ, pPKCθ, pPKCε, and pPKCη were additionally incubated at 25 °C in the absence or presence of MMS 0.04% for 1 h as a control condition. The activation of Rad53 was analyzed by Western blot. (**C**) Serial dilutions of exponentially growing cultures of the *tetO_7_:PKC1* (JCY1471) strain transformed with the same plasmids were spotted onto YPD plates and YPD plates supplemented with doxycycline 10 μg/mL and incubated at 25 °C for 3 days.

**Figure 4 ijms-24-15796-f004:**
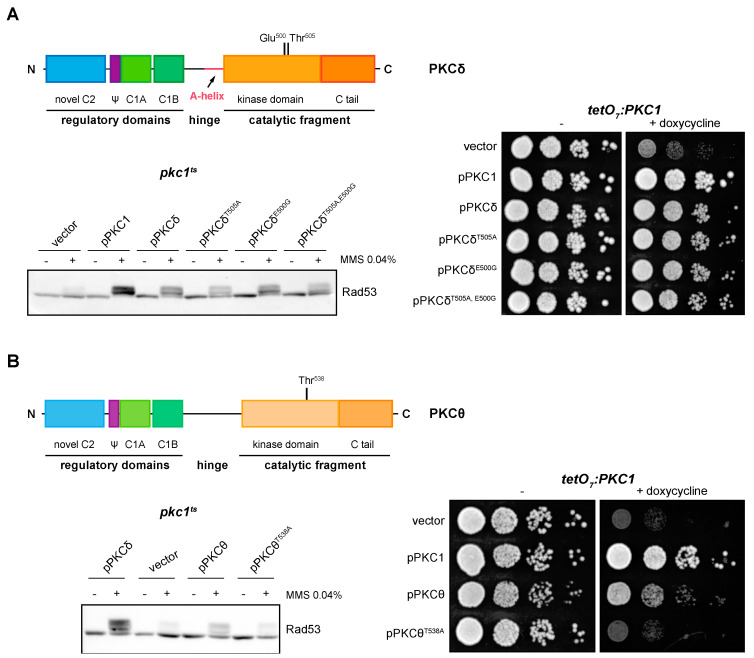
Suppression of the DNA integrity checkpoint activation and growth defects of *pkc1* mutant strains by mutants of PKCδ and pPKCθ in activation loop phosphorylation. (**A**) *Left upper panel:* schematic representation of the PKCδ protein indicating the residues of the activation loop that were mutated. *Left bottom panel:* exponentially growing cultures of the *pkc1^ts^* (JC6-3a) strain transformed with plasmids pPKC1, pPKCδ, pPKCδ^T505A^, pPKCδ^E500G^, pPKCδ^T505A,E500G^ or an empty vector were transferred to 37 °C for 3 h and then incubated in the absence or presence of MMS 0.04% for 1 h. The activation of Rad53 was analyzed by Western blot. *Right panel*: serial dilutions of exponentially growing cultures of the *tetO_7_:PKC1* (JCY1471) strain transformed with the same plasmids were spotted onto YPD plates and YPD plates supplemented with doxycycline 10 μg/mL and incubated at 25 °C for 3 days. (**B**) *Left, upper panel:* schematic representation of the PKCθ protein indicating the residue of the activation loop that was mutated. *Left, bottom panel:* exponentially growing cultures of the *pkc1^ts^* (JC6-3a) strain transformed with plasmids pPKCδ, pPKCθ, pPKCθ^T538A^ or an empty vector were transferred to 37 °C for 3 h and then incubated in the absence or presence of MMS 0.04% for 1 h. The activation of Rad53 was analyzed by Western blot. *Right panel:* serial dilutions of exponentially growing cultures of the *tetO_7_:PKC1* (JCY1471) strain transformed with the same plasmids were spotted onto YPD and YPD plates supplemented with doxycycline 10 μg/mL and incubated at 25 °C for 3 days.

**Figure 5 ijms-24-15796-f005:**
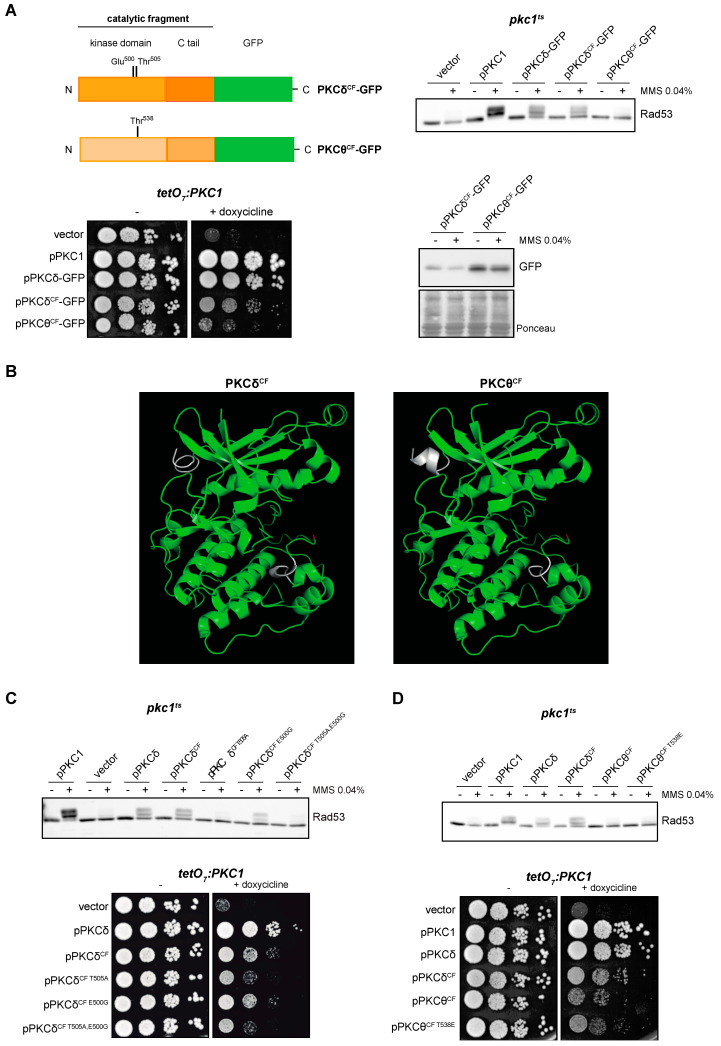
Suppression of the DNA integrity checkpoint activation and growth defects of *pkc1* mutant strains by the catalytic fragments of PKCδ and PKCθ. (**A**) *Left, upper panel*: schematic representation of the PKCδ and PKCθ catalytic fragments tagged with GFP indicating the residues of the activation loop that were mutated. *Right, upper panel*: exponentially growing cultures of the *pkc1^ts^* (JC6-3a) strain transformed with plasmids pPKC1, pPKCδ-GFP, pPKCδ^CF^-GFP, pPKCθ^CF^-GFP or an empty vector were transferred to 37 °C for 3 h and then incubated in the absence or presence of MMS 0.04% for 1 h. The activation of Rad53 was analyzed by Western blot. *Left, bottom pane*l: serial dilutions of exponentially growing cultures of the *tetO_7_:PKC1* (JCY1471) strain transformed with the same plasmids were spotted onto YPD and YPD plates supplemented with doxycycline 10 μg/mL and incubated at 25 °C for 3 days. *Right, bottom panel:* protein levels of PKCδ^CF^-GFP and PKCθ^CF^-GFP of the samples in A (first panel) were analyzed by Western blot. Ponceau staining of the membrane is shown as loading control. (**B**) Modeling of the tertiary structure of the catalytic fragments of PKCδ and PKCθ. The Thr residue of the activation loop is highlighted in red and the differences between the predicted structure of PKCδ^CF^ and PKCθ^CF^ are highlighted in white. (**C**) *Upper panel:* exponentially growing cultures of the *pkc1^ts^* (JC6-3a) strain transformed with plasmids pPKC1, pPKCδ, pPKCδ^CF^-GFP, pPKCδ^CF T505A^-GFP, pPKCδ^CF E500G^-GFP, pPKCδ^CF T505A,E500G^-GFP or an empty vector were transferred to 37 °C for 3 h and then incubated in the absence or presence of MMS 0.04% for 1 h. The activation of Rad53 was analyzed by Western blot. *Bottom panel:* serial dilutions of exponentially growing cultures of the *tetO_7_:PKC1* (JCY1471) strain transformed with the same plasmids were spotted onto YPD plates and YPD plates supplemented with doxycycline 10 μg/mL and incubated at 25 °C for 3 days. (**D**) *Upper panel:* exponentially growing cultures of the *pkc1^ts^* (JC6-3a) strain transformed with plasmids pPKC1, pPKCδ-GFP, pPKCδ^CF^-GFP, pPKCθ^CF^-GFP, pPKCθ^CF T538E^-GFP or an empty vector were transferred to 37 °C for 3 h and then incubated in the absence or presence of MMS 0.04% for 1 h. The activation of Rad53 was analyzed by Western blot. *Bottom panel:* serial dilutions of exponentially growing cultures of the *tetO_7_:PKC1* (JCY1471) strain transformed with the same plasmids were spotted onto YPD and YPD plates supplemented with doxycycline 10 μg/mL and incubated at 25 °C for 3 days.

**Figure 6 ijms-24-15796-f006:**
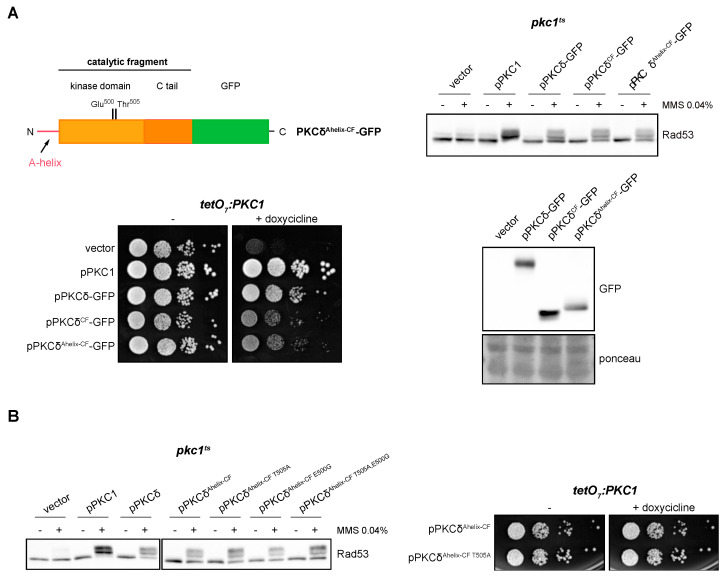
Suppression of the activation of the DNA integrity checkpoint and growth defects of *pkc1* mutant strains by the catalytic fragment of PKCδ with the A-helix. (**A**) *Left, upper panel:* schematic representation of the PKCδ catalytic fragment tagged with GFP with the addition of the A-helix in N-terminal indicating the residues of the activation loop that were mutated. *Right, upper panel:* exponentially growing cultures of the *pkc1^ts^* (JC6-3a) strain transformed with plasmids pPKC1, pPKCδ-GFP, pPKCδ^CF^-GFP, pPKCδ^Ahelix-CF^-GFP or an empty vector were transferred to 37 °C for 3 h and then incubated in the absence or presence of MMS 0.04% for 1 h. The activation of Rad53 was analyzed by Western blot. *Left, bottom panel:* serial dilutions of exponentially growing cultures of the *tetO_7_:PKC1* (JCY1471) strain transformed with the same plasmids were spotted onto YPD plates and YPD plates supplemented with doxycycline 10 μg/mL and incubated at 25 °C for 3 days. *Right, bottom panel:* Exponentially growing cultures of the *pkc1^ts^* (JC6-3a) strain transformed with plasmids pPKCδ-GFP, pPKCδ^CF^-GFP, pPKCδ^Ahelix-CF^-GFP or with an empty vector were incubated at 37 °C for 3 h. Protein levels of the different versions of PKCδ tagged with GFP were detected by Western blot. Ponceau staining of the membrane is shown as loading control. (**B**) *Left panel:* exponentially growing cultures of the *pkc1^ts^* (JC6-3a) strain transformed with plasmids pPKC1, pPKCδ, pPKCδ^Ahelix-CF^-GFP, pPKCδ^Ahelix-CF T505A^-GFP, pPKCδ^Ahelix-CF E500G^-GFP, pPKCδ^Ahelix-CF T505A,E500G^-GFP or with an empty vector were transferred to 37 °C for 3 h and then incubated in the absence or presence of MMS 0.04% for 1 h. The activation of Rad53 was analyzed by Western blot. *Right panel:* serial dilutions of exponentially growing cultures of the *tetO_7_:PKC1* (JCY1471) strain transformed with plasmids pPKCδ^Ahelix-CF^-GFP or pPKCδ^Ahelix-CF T505A^-GFP were spotted onto YPD and YPD plates supplemented with doxycycline 10 μg/mL and incubated at 25 °C for 3 days.

**Figure 7 ijms-24-15796-f007:**
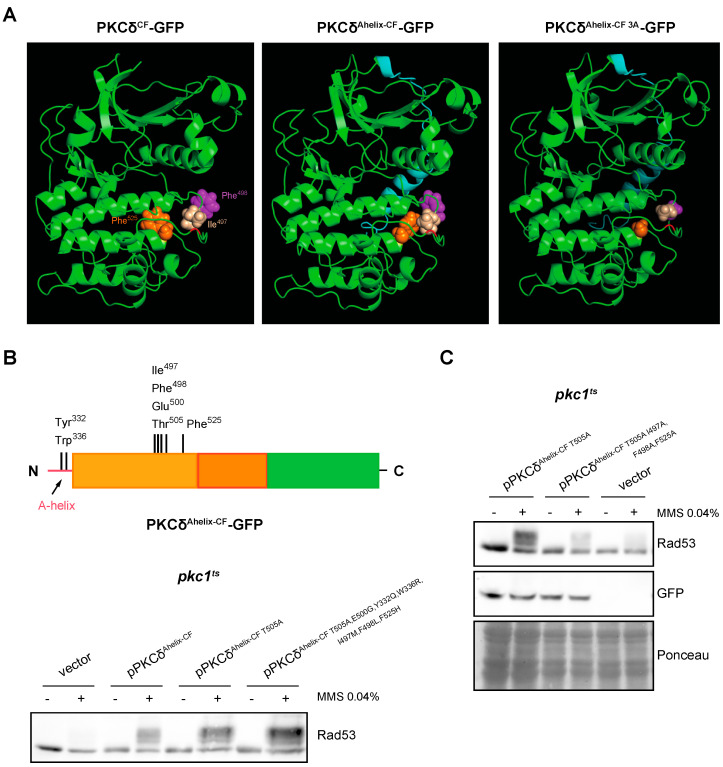
Analysis of DNA integrity checkpoint activation by PKCδ catalytic fragment with the A-helix mutated in key residues. (**A**) Modeling of the effect of the addition of the A-helix on the tertiary structure of the catalytic fragment of PKCδ and PKCδ^Ahelix-CF I497A,F498A,F525A^ (PKCδ^Ahelix-CF 3A^). The Thr residue of the activation loop is highlighted in red and the A-helix in blue. The side chains of the residues Ile^497^ (beige), Ile^498^ (purple), and Phe^525^ (orange) are represented with sphere structure. (**B**) *Upper panel:* schematic representation of the PKCδ catalytic fragment tagged with GFP with the addition of the A-helix in N-terminal indicating the residues that were mutated. *Bottom panel:* exponentially growing cultures of the *pkc1^ts^* (JC6-3a) strain transformed with plasmids pPKCδ^Ahelix-CF T505A^-GFP, pPKCδ^Ahelix-CF T505A,E500G,Y332Q,W336R,I497M,F498L,F525H^-GFP or an empty vector were transferred to 37 °C for 3 h and then incubated in the absence or presence of MMS 0.04% for 1 h. The activation of Rad53 was analyzed by Western blot. (**C**) Exponentially growing cultures of the *pkc1^ts^* (JC6-3a) strain transformed with plasmids pPKCδ^Ahelix-CF T505A^-GFP, pPKCδ^Ahelix-CF T505A,I497A,F498A,F525A^-GFP or an empty vector were transferred to 37 °C for 3 h and then incubated in the absence or presence of MMS 0.04% for 1 h. The activation of the Rad53 protein and protein levels of PKCδ^Ahelix-CF^ mutant versions tagged with GFP were analyzed by Western blot.

**Figure 8 ijms-24-15796-f008:**
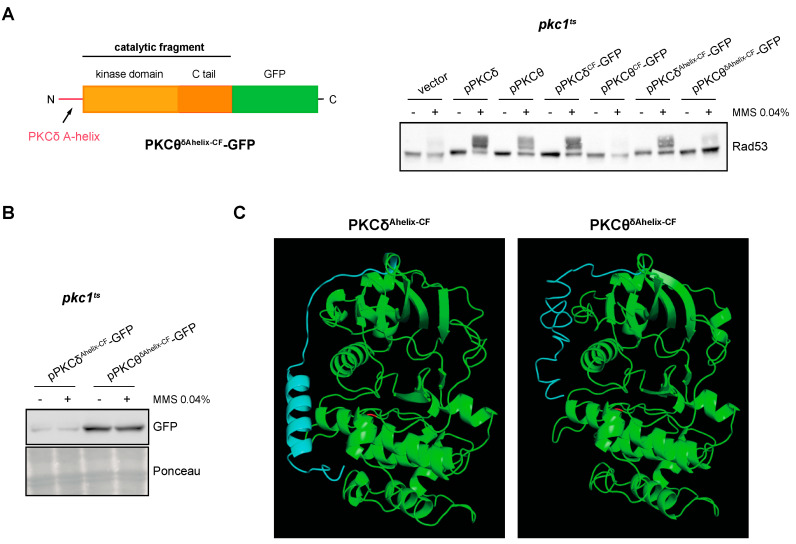
Analysis of DNA integrity checkpoint activation by the PKCθ catalytic fragment with PKCδ A-helix. (**A**) *Left panel:* schematic representation of the chimeric protein containing the A-helix of PKCδ fused to the N-terminal end of the PKCθ catalytic fragment tagged with GFP. *Right panel:* exponentially growing cultures of the *pkc1^ts^* (JC6-3a) strain transformed with plasmids pPKCδ, pPKCθ, pPKCδ^CF^-GFP, pPKCθ^CF^-GFP, pPKCδ^Ahelix-CF^-GFP, pPKCθ^δAhelix-CF^-GFP or an empty vector were transferred to 37 °C for 3 h and then incubated in the absence or presence of MMS 0.04% for 1 h. The activation of Rad53 was analyzed by Western blot. (**B**) PKCδ^Ahelix-CF^-GFP and PKCθ^δAhelix-CF^-GFP levels were detected by Western blot. Ponceau staining of the membrane is shown as loading control. (**C**) Modeling of the tertiary structure of PKCδ^Ahelix-CF^ and PKCθ^δAhelix-CF^. The Thr residue of the activation loop is highlighted in red and the PKCδ A-helix sequence in blue.

## Data Availability

Data is contained within the article or Appendix A.

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
