# Peer review of "A Multimodel Study of the Role of Novel PKC Isoforms in the DNA Integrity Checkpoint"

_ijms, 2023, doi:10.3390/ijms242115796_

Round 1

Reviewer 1 Report

Comments and Suggestions for Authors

Comments on the Quality of English Language

Minor editing

Reviewer 2 Report

Comments and Suggestions for Authors

The manuscript entitled “A multimodel study of the role of novel PKC isoforms in the DNA integrity checkpoint” investigated the role of PKC isoforms in the control of DNA integrity checkpoint activation. It was first found that PKCd is involved in activating the checkpoint effector kinase CHK1 in mouse embryonic stem cells. Further analysis with the yeast model revealed that among all PKC isoforms, only PKCθ exhibited regulatory function for checkpoint effector Rad53 and alleviated growth suppression of the pkc1ts strain in addition to PKCd. By constructing plasmids that contain different fragments of the kinases, it was demonstrated that the catalytic fragment of PKCδ is sufficient for its activation. On the other hand, the A-helix of the kinase may provide an alternative activation mechanism that is independent of the phosphorylation of the activation loop of PKCd; while the activation of PKCθ requires the phosphorylation of Thr538. The study provides interesting and useful information on the specific kinds of PKC involved in the regulation of DNA integrity checkpoint control. The manuscript is in general well-written. Some comments for consideration are listed below:

1) The title of the manuscript that indicates the current study as a “multimodel study” seems to be misleading. The mammalian stem cell system was only utilized to check the effect of PKCd knock-out on the activation of CHK1; while the rest of the studies were totally based on the yeast system. The authors should consider validating some of their findings, for example, the unique role of the A-helix in the activation of PKCd, the regulatory role of PKCθ for DNA integrity checkpoint control, etc. in the mESC system.   

2) It was shown in Figure 3 that the over-expression of PKC isoforms other than PKCd and θ failed to activate Rad53 and suppress the growth defect of the mutant strain tetO7:PKC1. Whether all these PKCs were successfully over-expressed in the yeast system and that their protein expression level was comparable to those of PKCd and θ should be evaluated.

3) The loading controls are quite inconsistent throughout the manuscript. Sometimes it is ponceau S (Fig. 1&5&7); sometimes it is Cdc28; and in most cases, it is totally missing. The loading controls should be provided and unified to better judge the relative protein level of different samples.

4) The description of repeat experiments is quite confusing in the figure legend of Fig. 1: “A total of 8 independent experiments were carried out, including cells from 3 independent clones for Pkcδ+/+ and 7 independent clones for Pkcδ-/- cells, resulting in a total of 16 control samples and 22 Pkcδ-/- samples.” Please clarify.

5) According to figure legend in Fig5B, “The Thr residue of the activation loop is highlighted in red..”. However, there is nothing labeled in red in the figure.     

Round 2

Reviewer 1 Report

Comments and Suggestions for Authors

In this revised version of the manuscripts the authors addressed most of my concerns and add crucial controls, therefore increasing the significance of their results. Therefore, I support the publication of the manuscript in ijms.

Reviewer 2 Report

Comments and Suggestions for Authors

The authors have addressed the issues raised.